# The Effects of Different Inlet Configurations on Particles Transport and Residence Time in a Shallow and Narrow Coastal Lagoon: A Numerical Based Investigation

**Zuraini Zainol and Mohd Fadzil Akhir ***

Institute of Oceanography and Environment, Universiti Malaysia Terengganu, Kuala Terengganu 21030, Malaysia; zuraini.z@umt.edu.my
* Correspondence: mfadzil@umt.edu.my

**Abstract:** Setiu Lagoon experienced shoreline alterations, leading to the opening of a new inlet and the closing of the old one. To evaluate the effects of these alterations on the tidal flow in this shallow and narrow lagoon, a numerical hydrodynamic model was developed. The model-predicted water level and current velocities were validated against field measurements, which yielded a high simulation skill. The transport of particles and residence time under different inlet configurations were also investigated through the implementation of numerical tracers released from R1, R2, and R3, which represent different pollution sources. The results indicated that the particle transport and residence time in Setiu Lagoon varied spatially and temporally depending on the release locations, proximity to the inlet, and tidal current strength. Comparing the scenarios, the flushing efficiency in Setiu Lagoon was improved with the presence of both inlets, resulting in a shorter residence time of less than 50 days. The findings of this research are vital for understanding the water current flow and residence time in this restricted lagoon, while evaluating the possible adverse effects on its water quality. Although this work is case-study based, the output is applicable to other lagoon systems with similar characteristics.

**Keywords:** residence time; particles transport; tidal currents; different inlet configurations; Setiu Lagoon





## 1. Introduction

Inlets are common coastal features connecting estuaries, lagoons, and bays with the open ocean. They play a great role as a gateway to the sea, providing critical habitat for wildlife and sustaining ecological diversity [1]. Inlets are also crucial for shore and dune processes and the exchange of both water and sediments, in which their dynamics are mainly controlled by complex interactions between different environmental drivers (i.e., wind, tides, and waves) and their own topographic features [2]. However, in recent years, inlets are facing overwhelming sedimentary balance degradation, in which the ingress of sediments into these entrances has caused constriction that would eventually lead to a partial or full blockage of the system [3,4]. These alterations can be severe, and result in the formation of poorly ventilated basins with long flushing time, which would degrade the water quality [5,6].

Globally, sedimentation in tidal inlets is one of the major problems faced by many lagoon ecosystems [7,8]. The best known case would be the multi inlet system of Ria Formosa lagoon, Southern Portugal, which was undergoing infilling processes and was artificially relocated in 1997 to improve its hydrodynamic efficiency [9–12]. Another important example is the siltation problem experienced by the Punta Umbría inlet in Ría de Huelva estuary, Spain, that requires intense dredging works to avoid the closure of the inlet [2]. Within the Southeast Asia region, the Vietnamese government has built a 400 m long jetty to deal with the increase sedimentation problems in the inlet of the De Gi estuary [13]. Meanwhile in Malaysia, excess sediment accretion has caused a serious problem in Salut-Mengkabong

Lagoon, Sabah, where the main inlet is becoming narrower, and certain areas are becoming shallower, which poses a threat to water quality and nearshore marine habitats [14].

Setiu Lagoon is no exception regarding the sedimentation issue. The time-lapse images digitized from Google Earth have shown the position of the inlet keeps changing over time, and there is a natural relocation of the river mouth through a closing of an old inlet and the opening of a new one (Figure 1). Previous study by Kasawani et al. [15] suggested that the erosion, accretion, and transportation of sediment resulted from the strong actions of monsoonal wave, wind, and current are the factors responsible for these changes. Although irregular, this inlet shift is concerning, since previous literatures have documented hydrodynamic and ecological changes resulted from this alteration [16–18]. Furthermore, it has been suggested that any natural or anthropogenic modifications to the tidal inlet can disturb the hydro-morphological balance, whether in the short- or long-term, and a long water residence time may cause the lagoon system to be in eutrophic state [4,19], particularly during the closed phase.

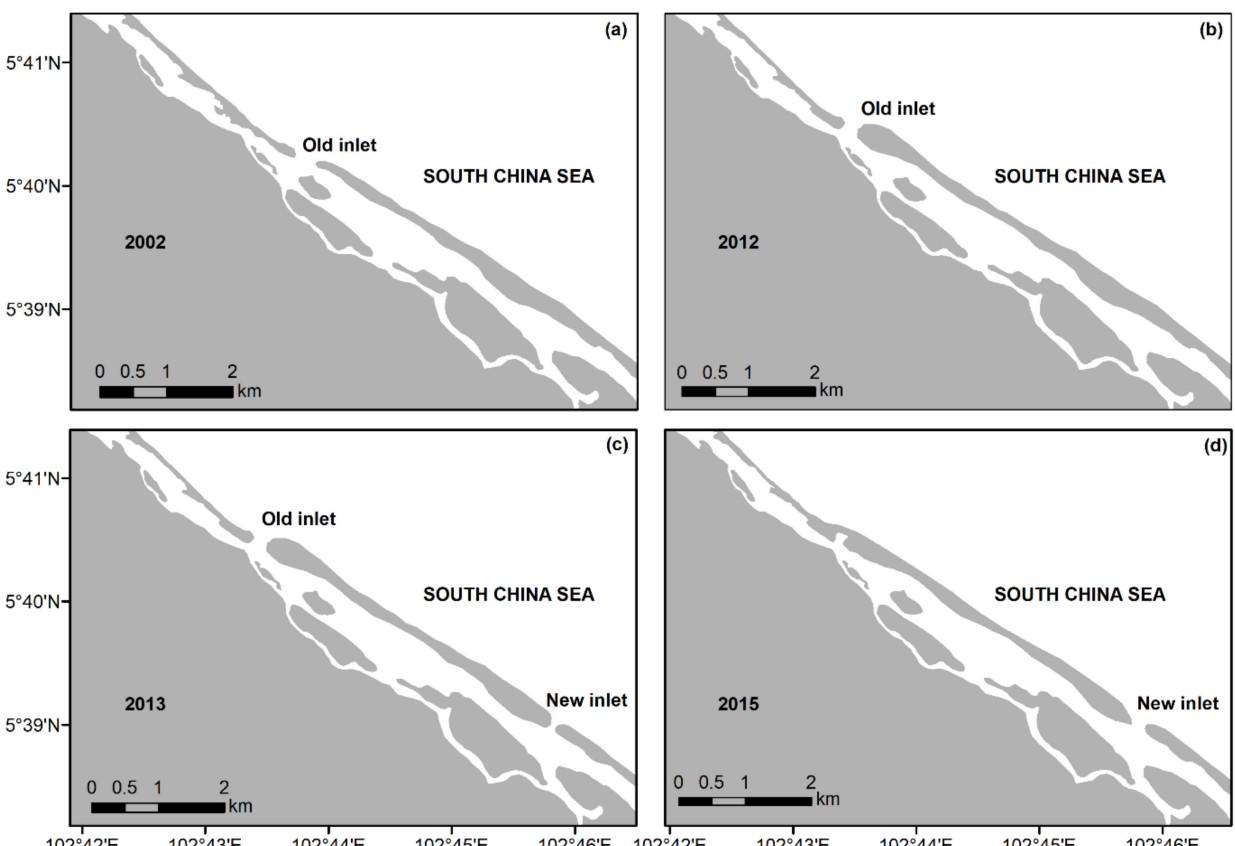

**Figure 1.** The shoreline changes in Setiu Lagoon during (**a**) 2002, (**b**) 2012, (**c**) 2013, and (**d**) 2015, digitized from Google Earth.

Since it is important to monitor the potential impact of modifications on natural water bodies, this paper attempts to evaluate the effects of inlet relocation in Setiu Lagoon by performing numerical simulations of Setiu Wetland for the past and present configurations, which has not been addressed in previous studies. A coupled hydrodynamic and particle tracking model was applied to assess the efficiency of the inlet states (old inlet, both inlets, and new inlet) in removing the tracers out of the Setiu Lagoon under the influence of water current driven by tides and winds. The tracer transport analysis was then made with a specific objective of evaluating the residence time of all three conditions. Finally, the effects of the inlet relocation in terms of water renewal are discussed.

## 2. Methodology

### 2.1. Study Site

Setiu Lagoon (Terengganu, Malaysia) is situated within the Setiu Wetland, which is known as the largest natural wetland in the east coast of Peninsular Malaysia (ECPM), and has been listed as one of 17 priority conservation sites in the Malaysian Wetland Directory for its ecological and conservation values [16,18,20,21]. This 14 km-length lagoon has shallow water depths ranging between 0.3 and 3.2 m [16]. Setiu Lagoon is also unique due to its narrow channels attributable to the presence of vegetated sand islands within the lagoon (Figure 2). Freshwater discharge into the lagoon comes predominantly from the Setiu River, which flows directly into the lagoon and Berambak Lake, connected through the Ular River (Figure 2). Upstream of Setiu River, there is the urban area of Penarik town, which contributes to the domestic wastes input in this lagoon (Figure 2) [17,18,22,23].

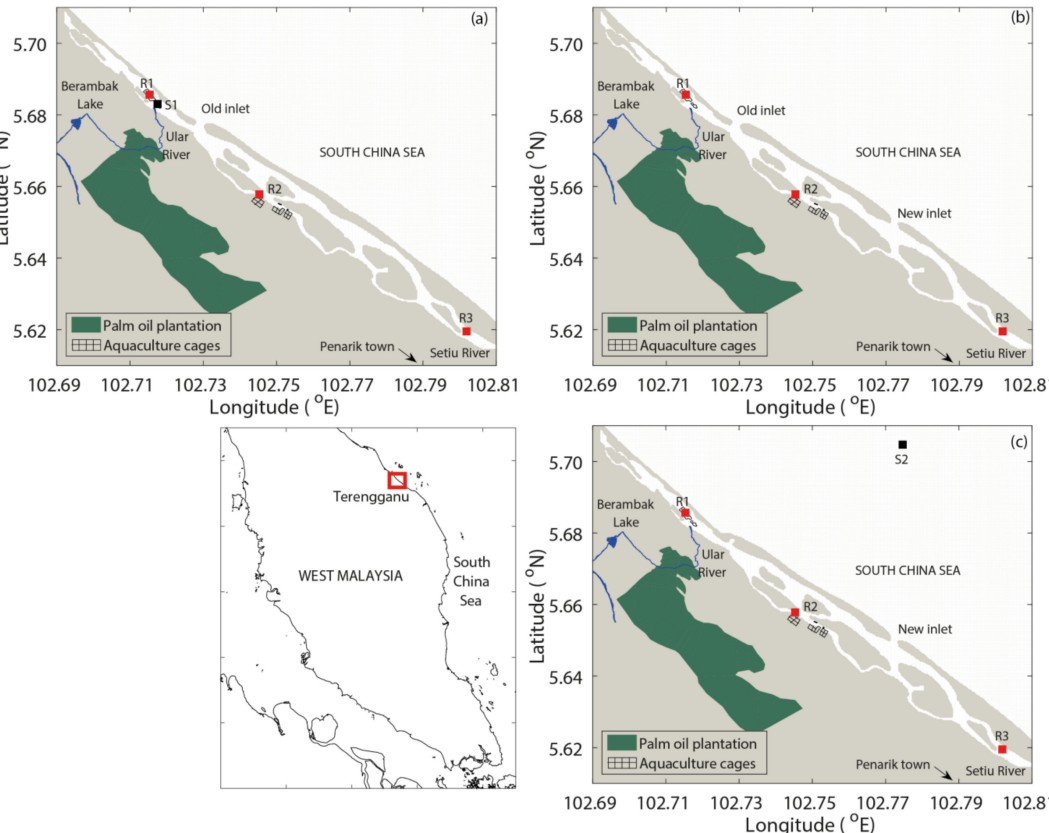

**Figure 2.** Map of Setiu Wetland showing a narrow lagoon, which is located in the state of Terengganu, Malaysia. The lagoon receives freshwater from the Setiu River and Berambak Lake (connected through the Ular River). The green patch indicates the palm oil plantation area, while the grids represent the aquaculture cages. S1 indicates the current meter deployment position in 2015, and S2 indicates the Acoustic Wave and Current (AWAC) profiler deployment location in 2020. R1, R2, and R3 indicate the release points for the tracers. The study area refers to three different inlet configurations, namely (**a**) old inlet, (**b**) both inlets, and (**c**) new inlets, for the interpretation of the results.

Although the total surface area of Setiu Lagoon is only 880 ha [24], it holds an economic value in provisioning aquatic resources for the livelihoods of the local residents. According to Jamilah et al. [25], the direct and indirect (i.e., coastline protection, carbon sequestration, and offshore fishery linkages) use-value of the mangroves in this area was estimated to be around RM 1452–10,700 per hectare (RM: currency in Malaysian Ringgits). Moreover, aquaculture activities such as oyster farming, brackish water cage culture, pond culture, and pen culture are growing rapidly within the lagoon [17,22]. Since it was introduced in the

late 1970s, the production of fish cage culture in Setiu Lagoon has increased tremendously from 0.88 metric tons in 1995, to 1460.17 metric tons in 2012 [26]. Meanwhile, the upstream area of this lagoon is dominated by agriculture activities, primarily palm oil cultivation, where a vast expansion of this activity has been documented between 2008 and 2011 [16].

Setiu Lagoon is a mesotidal system with a semi-diurnal tidal domination, and a tidal range variation of between 1.0 and 1.5 m [16,27]. In terms of climate variability, Setiu Lagoon is subjected to two monsoons (i.e., northeast and southwest) and two inter-monsoon periods. The northeast monsoon with heavy rainfalls usually prevails from November to March, whereas the southwest monsoon with dry conditions lasts from May to September [28–30]. Generally, the surface water current in the ECPM flows parallel to the coastline; this means in a southward direction during the northeast monsoon, and northward during the southwest monsoon [28,31,32].

Previously, Setiu Lagoon was connected to the South China Sea through an inlet located at the northern part of the lagoon (hereafter referred to as old inlet; Figure 2a). This old inlet is situated downstream of the Ular River and near aquaculture farms (Figure 2a). Observation from Google Earth has shown that a new inlet has formed (Figure 2b,c), which is located approximately 4 km south of the old one. Over time, sediments were observed to accumulate at the entrance of the old inlet that finally led to the closing of the river mouth (Figure 2c).

## 2.2. In Situ Data Collection

To validate the hydrodynamic simulation results, a unit of Valeport® current meter (Model 308, UK) with 5-min data recording intervals was moored at S1 for one week (21–28 October 2015; Figure 2a). Meanwhile, a unit of Acoustic Wave and Current (AWAC) profiler was moored from 6–26 October 2020 at S2 with 10-min data recording intervals (Figure 2c). The bathymetry of the entire lagoon, as used in the hydrodynamic model, were collected in three phases (June 2014, November 2015, and August 2017) by using a Garmin® GPSmap echosounder (Model 298, UK) and SeaSTAR 3200 LR12 DGPS Receiver (The Netherlands).

## 2.3. Numerical Model Description

In order to study the transport distribution and residence time of particles in Setiu Lagoon, a two-dimensional horizontal surface hydrodynamic model of Setiu Wetland was implemented using the Delft3D FLOW module to undertake experimental scenarios under three different inlet configurations. It is noteworthy that Delft3D was successfully used by the authors to study the pollutant transport in Setiu Lagoon with the same setup conditions [18]. Since the general setup for the hydrodynamic model of Setiu Wetland and its calibration has been provided in that previous study, only a brief description of the model construction will be provided here. In this current study, three different computational grids were used, corresponding to the past and current configurations of Setiu Lagoon. Several refinements were applied to the local grids of Setiu Lagoon due to narrow channels and the presence of vegetated islands, resulting in squared grid resolutions that ranged between 36 to 100 m. For the old inlet scenario, the model grid consisted of 78,723 computational elements. Meanwhile, for both inlets and new inlet scenarios, the grid comprised of 78, 798, and 78,715 computational elements, respectively. Moreover, on the case of refined mesh for the lagoon, the computational time step was reduced to 15 sec to prevent instability during the simulation. For the hydrodynamic simulation, it was forced by winds and tides, while the freshwater input were turned off due to data limitation. 13 tidal constituents ($M_2$, $S_2$, $N_2$, $K_2$, $K_1$, $O_1$, $P_1$, $Q_1$, MF, MM, $M_4$, $MS_4$, and $MN_4$) obtained from the tidal toolbox of the Delft Dashboard software were imposed on the ocean boundary. On the other hand, wind field data used in the simulation was acquired from the European Centre for Medium-range Weather Forecast (ECMWF). The hydrodynamic model in the Setiu Lagoon was initialized on 00:00 1 May 2019 until 00:00 1 October 2019, and 00:00 1 November 2019 until 00:00 1 April 2020, representing the southwest and northeast monsoons, respectively. Three

scenarios were implemented, in which Scenario 1 represents the old inlet configuration, Scenario 2 represents the configuration of both inlets, and Scenario 3 is the new inlet configuration. The simulation interval specified as 180 min and a ramp function of 7 days was used for warm-up period.

For the estimation of residence time and transport trajectories in Setiu Lagoon, the Delft3D particle tracking (PART) module was utilized. The output of the FLOW simulations served as the input for the PART model. Three release locations (R1, R2, and R3; Figure 2) were selected, since R1 and R2 were situated near aquaculture ponds and farms, while R3 was located downstream of Setiu River where there is domestic waste input from Penarik town (Figure 2). In the PART model, a conservative tracer comprising of 200 particles was simultaneously released at R1, R2 and R3. The particles were released at 03:00 10 May 2019 and 15:00 13 November 2019 during the ebbing of southwest and northeast monsoons, respectively. At each release location, the number of particles was noted and tracked daily for 130 days, and the time required to flush out 63% of the initial concentration evenly distributed at time zero was considered as the residence time [33–35].

## 3. Results

### 3.1. Accuracy of Hydrodynamic Model Simulations

To test the capability of the hydrodynamic model, a set of statistical parameters (e.g., mean, standard deviation, root mean square error (RMSE), correlation coefficient, and index of the agreement) were derived from 30-min interval observed and simulated data. A perfect agreement between the simulation-based results and the field-based observations produced an index of '1', whereas complete disagreement produced an index of '0'. The time series comparisons between measured and simulated water levels and u and v velocities of the water current in Figure 3 show that both amplitude and phase of the parameters were clearly reproduced, suggesting a good agreement between the data. Furthermore, the simulation skills for all parameters, represented by the index of agreement, are also high (exceeding 0.7), indicating a good calibration of the model (Tables 1 and 2).

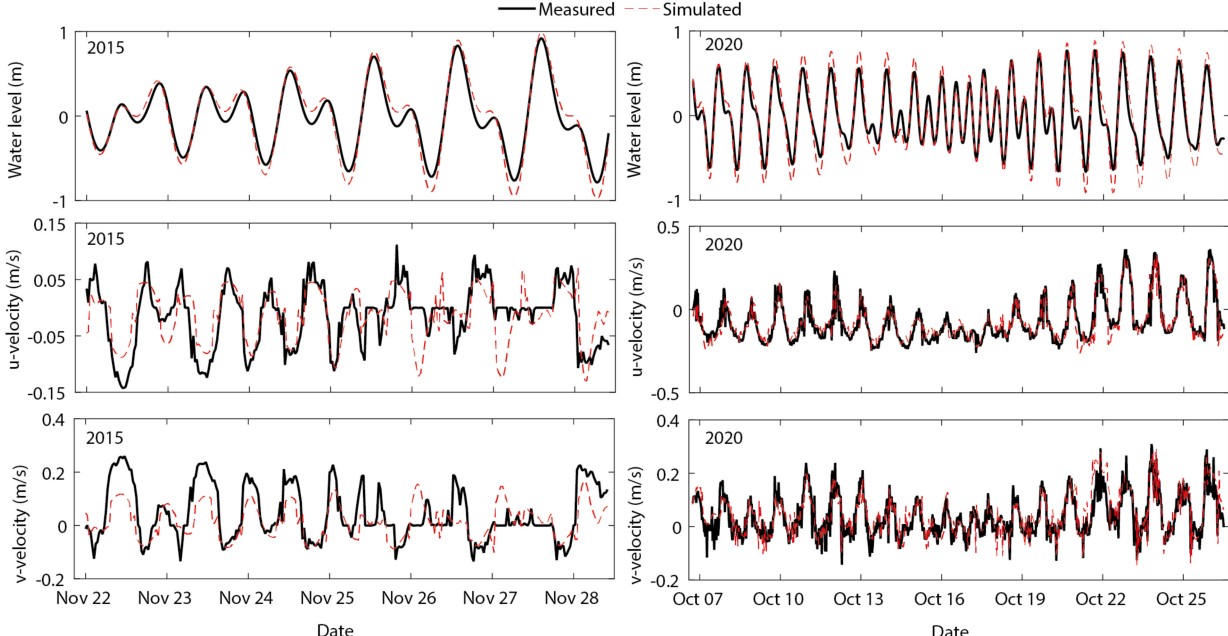

**Figure 3.** Time series comparison between measured and simulated water level and u and v velocities of the water current during 2015 (**left** panels) and 2020 (**right** panels). Please refer to Figure 2 for station locations.

**Table 1.** Statistical assessment of the hydrodynamic model performance at S1 for the water level, and u and v components of the water current.

| Parameter | Observation | | Model Simulation | | RMSE | Correlation Coefficient | Index of Agreement |
|---|---|---|---|---|---|---|---|
| | Mean | Std. Dev. | Mean | Std. Dev. | | | |
| Water level (m) | −0.03 | 0.38 | −0.02 | 0.46 | 0.11 | 0.98 | 0.98 |
| u-component of the water current (m/s) | −0.01 | 0.05 | −0.01 | 0.06 | 0.04 | 0.65 | 0.79 |
| v-component of the water current (m/s) | 0.04 | 0.10 | 0.02 | 0.06 | 0.08 | 0.67 | 0.73 |

**Table 2.** Statistical assessment of the hydrodynamic model performance at S2 for the water level, and u and v components of the water current.

| Parameter | Observation | | Model Simulation | | RMSE | Correlation Coefficient | Index of Agreement |
|---|---|---|---|---|---|---|---|
| | Mean | Std. Dev. | Mean | Std. Dev. | | | |
| Water level (m) | 0.00 | 0.35 | 0.00 | 0.46 | 0.14 | 0.97 | 0.97 |
| u-component of the water current (m/s) | −0.07 | 0.12 | −0.07 | 0.12 | 0.05 | 0.93 | 0.96 |
| v-component of the water current (m/s) | 0.04 | 0.08 | 0.05 | 0.08 | 0.05 | 0.83 | 0.90 |

*3.2. Water Current Velocity Simulation*

Since Setiu Lagoon is a tidally-dominated system, it is important to understand the water circulation pattern under different inlet configurations and seasonal influence prior to any result interpretation. Generally, the water current within Setiu Lagoon was found to be the strongest near the inlet area that reached up to 0.5 m/s (Figure 4). Further, it was apparent that the central part of the lagoon was associated with stronger current speed than the northern and southern parts in all scenarios (Figure 4). Under the scenario of the old inlet, which is located at the northern area of the lagoon, strong ebb currents were observed to flow from the southern part towards the river mouth (Figure 4a,c). Meanwhile, strong flood currents moved directly towards the central part of the lagoon (Figure 4b,d). Under the presence of both inlets, the tidal currents within the Setiu Lagoon were generally well distributed along the lagoon, could be due to similar energy dissipation (Figure 4e–h). On the other hand, due to the position of the new inlet that is located at the southern area of the Setiu Lagoon, the ebb tide currents propagated southward, from the central part towards the river mouth area, while the flood tides gushed towards the central sector (Figure 4i–l). At the coastal area, the water currents flowed parallel to the shoreline either in northward or southward directions (Figure 4).

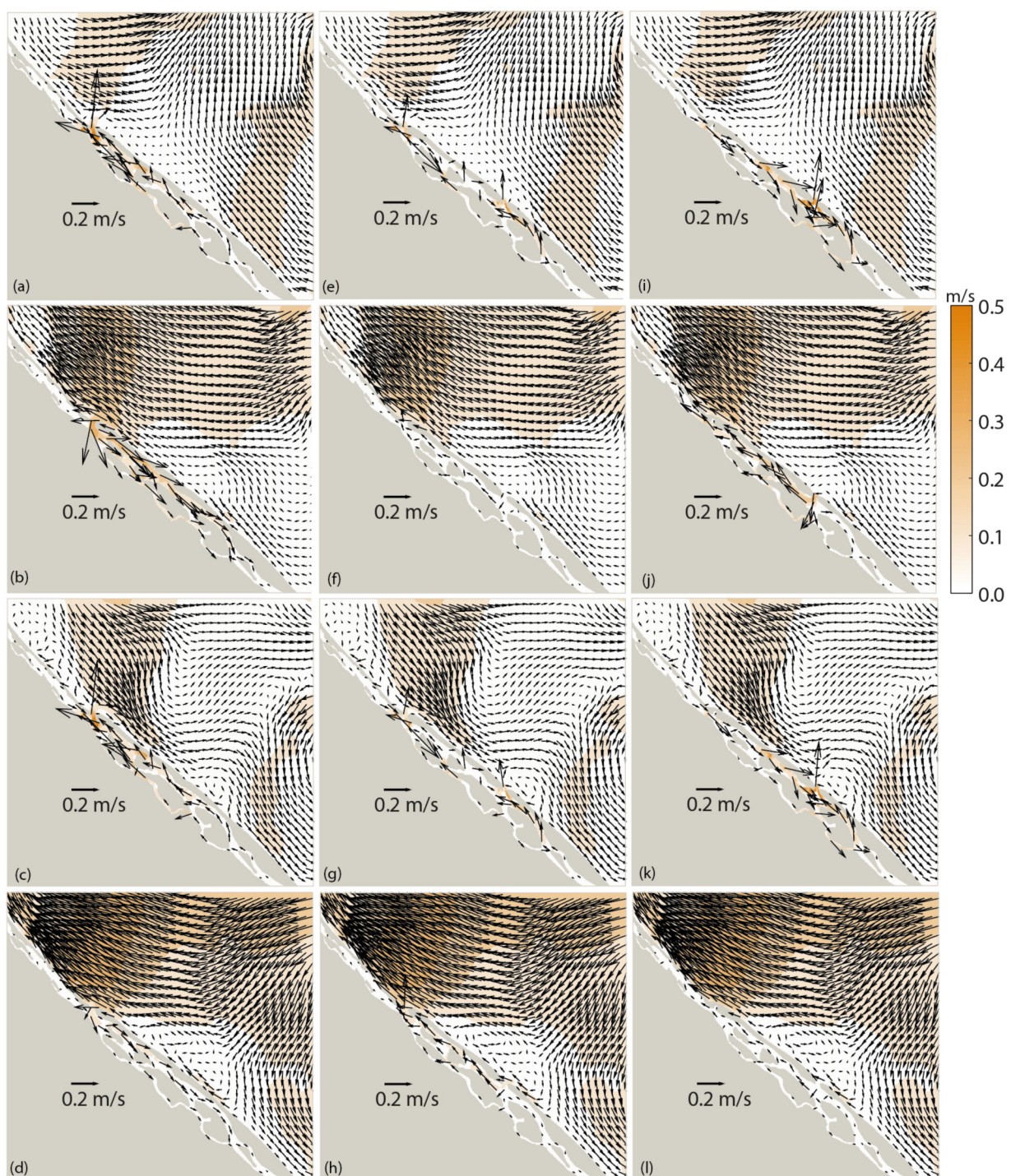

**Figure 4.** Distribution of water current in the Setiu Lagoon under old (**a**–**d**), both (**e**–**h**) and new (**i**–**l**) inlets configurations during the ebbing (**a** + **c**, **e** + **g**, **i** + **k**) and flooding (**b** + **d**, **f** + **h**, **j** + **l**) of the southwest (top two panels) and northeast (bottom two panels) monsoons.

### 3.3. Particle Trajectories under Different Inlet Configurations

Figures 5 and 6 show the distribution of particles that are seeded at R1, R2 and R3 with the old inlet configuration at varying time frames of the southwest and northeast monsoons, respectively. From these figures, it could be clearly observed that the particles movement during the southwest and northeast monsoons was faster in the northern than southern parts, evident from a large number of retained particles at the southern area of

Setiu Lagoon from Day 30 onwards (Figures 5 and 6). Similar results were documented for the particles simulated under both inlets configuration, in which the excursion of particles in the northern part of the Setiu Lagoon was faster, since most of the trapped particles were concentrated at the southern part of the lagoon (Figures 7 and 8). However, in the simulation for the new inlet configuration, a different result was obtained, in which the transport of particles towards the river mouth was faster in the southern part, with a majority of trapped particles was retained in the northern area of the lagoon (Figures 9 and 10). Comparison between different monsoon conditions reveals that the northeast monsoon expedited the transport of particles out of the Setiu Lagoon, demonstrated by larger number of exited particles on Day 120 than the southwest monsoon (Table 3). In terms of the inlet configurations, the both inlets scenario was found to be the most efficient in removing particles by flushing out the largest number of the tracers released (Table 3). In addition, one of the common characteristics observed from all the scenarios was that all the exited particles were transported either to the north or south, following the general flow of the coastal current, and eventually, some of them exited Setiu Wetland domain permanently.

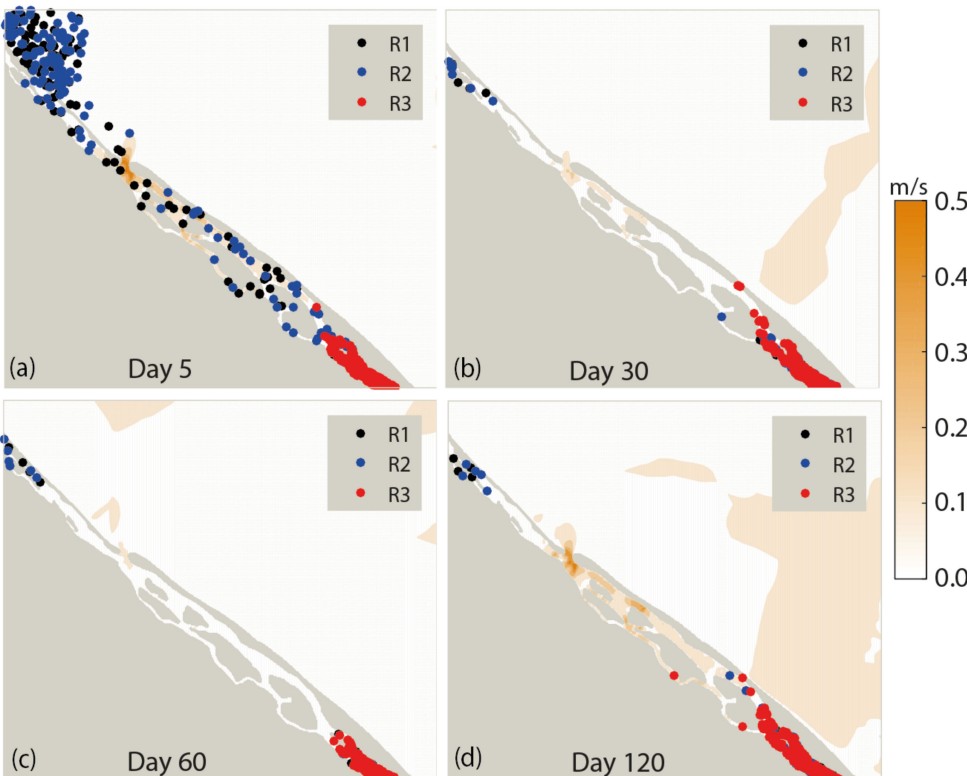

**Figure 5.** Spatial distributions of the current speed and trajectories for particles released at R1, R2 and R3 (please refer Figure 2 for initial release locations) on (**a**) Day 5, (**b**) Day 30, (**c**) Day 60 and (**d**) Day 120 during the southwest monsoon for old inlet configuration.

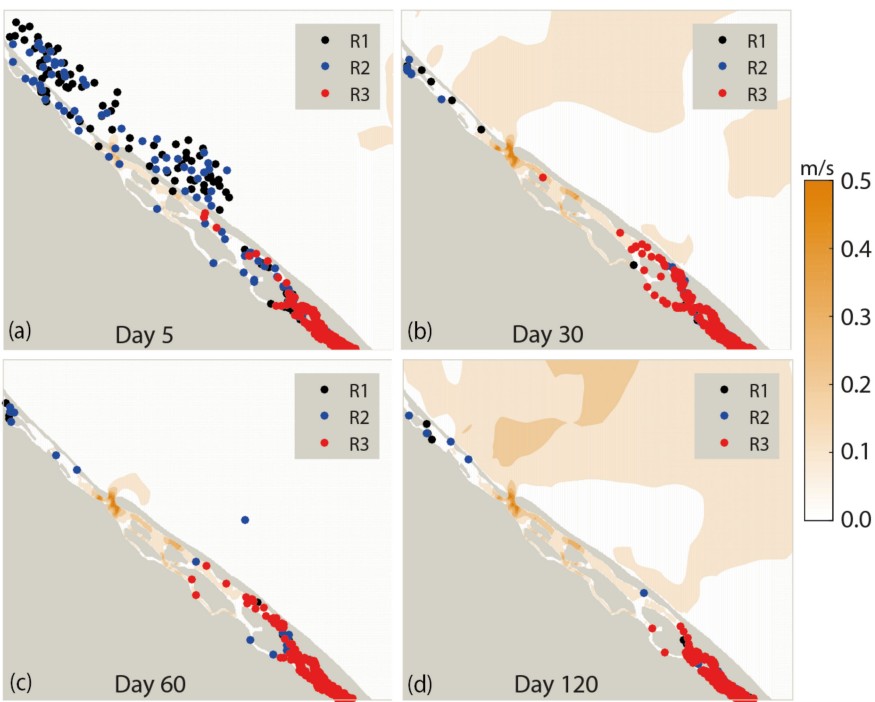

**Figure 6.** Spatial distributions of the current speed and trajectories for particles released at R1, R2 and R3 (please refer Figure 2 for initial release locations) on (**a**) Day 5, (**b**) Day 30, (**c**) Day 60 and (**d**) Day 120 during the northeast monsoon for old inlet configuration.

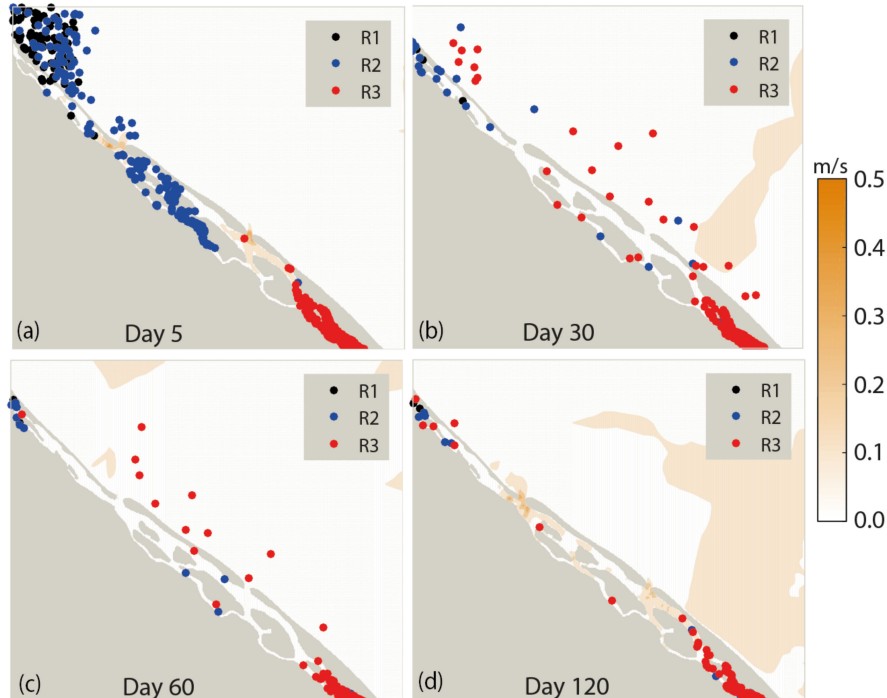

**Figure 7.** Spatial distributions of the current speed and trajectories for particles released at R1, R2 and R3 (please refer Figure 2 for initial release locations) on (**a**) Day 5, (**b**) Day 30, (**c**) Day 60 and (**d**) Day 120 during the southwest monsoon for both inlets configuration.

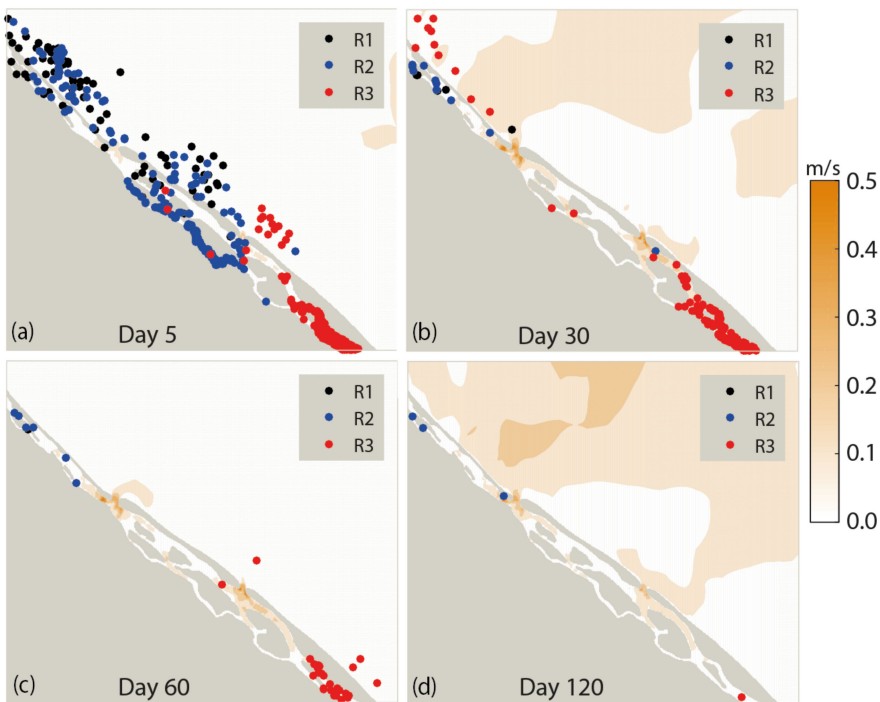

**Figure 8.** Spatial distributions of the current speed and trajectories for particles released at R1, R2 and R3 (please refer Figure 2 for initial release locations) on (**a**) Day 5, (**b**) Day 30, (**c**) Day 60 and (**d**) Day 120 during the northeast monsoon for both inlets configuration.

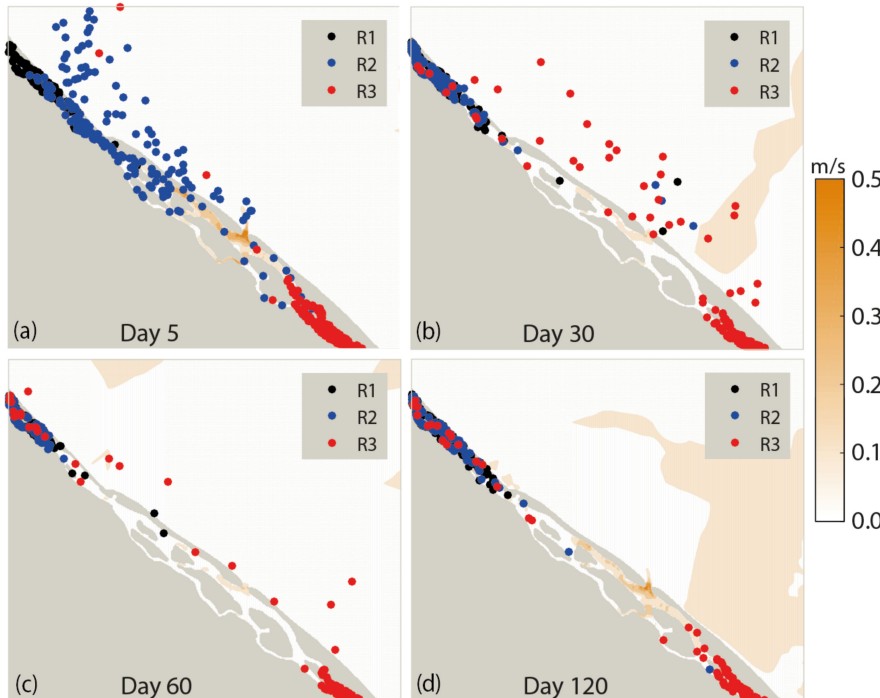

**Figure 9.** Spatial distributions of the current speed and trajectories for particles released at R1, R2 and R3 (please refer Figure 2 for initial release locations) on (**a**) Day 5, (**b**) Day 30, (**c**) Day 60 and (**d**) Day 120 during the southwest monsoon for new inlet configuration.

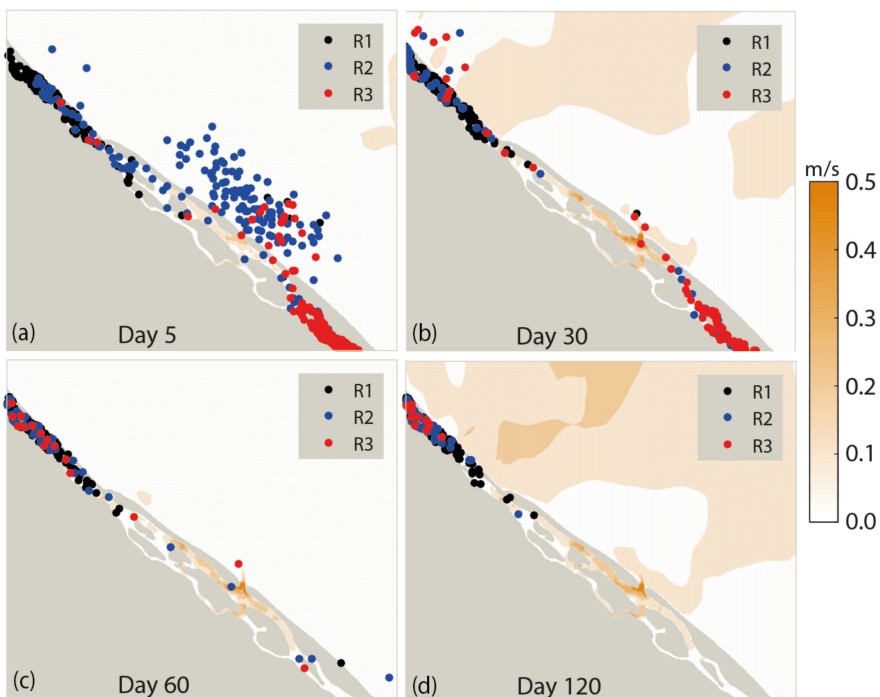

**Figure 10.** Spatial distributions of the current speed and trajectories for particles released at R1, R2 and R3 (please refer Figure 2 for initial release locations) on (**a**) Day 5, (**b**) Day 30, (**c**) Day 60 and (**d**) Day 120 during the northeast monsoon for new inlet configuration.

**Table 3.** The number of remaining particles released at R1, R2 and R3 on Days 5, 30, 60 and 120 of simulation. 'SW' denotes the southwest and 'NE'.

| Inlet Configuration | Release Point | Monsoon | Day 5 | Day 30 | Day 60 | Day 120 |
|---|---|---|---|---|---|---|
| | R1 | SW | 36 | 28 | 27 | 27 |
| | | NE | 30 | 26 | 25 | 24 |
| Old inlet | R2 | SW | 52 | 47 | 47 | 46 |
| | | NE | 45 | 38 | 36 | 33 |
| | R3 | SW | 200 | 200 | 199 | 198 |
| | | NE | 199 | 196 | 195 | 179 |
| | R1 | SW | 5 | 3 | 2 | 2 |
| | | NE | 24 | 4 | 1 | 0 |
| Both inlets | R2 | SW | 125 | 25 | 19 | 12 |
| | | NE | 123 | 11 | 6 | 3 |
| | R3 | SW | 200 | 172 | 142 | 101 |
| | | NE | 182 | 106 | 23 | 1 |
| | R1 | SW | 200 | 194 | 190 | 187 |
| | | NE | 193 | 186 | 175 | 160 |
| New inlet | R2 | SW | 134 | 106 | 103 | 101 |
| | | NE | 81 | 52 | 37 | 34 |
| | R3 | SW | 197 | 137 | 109 | 73 |
| | | NE | 178 | 64 | 15 | 13 |

*3.4. Residence Time in Setiu Lagoon*

Figure 11a,b display the estimation of residence time at three release locations under different inlet configurations during the southwest and northeast monsoons, respectively. In general, some scenarios have failed to estimate the residence time of particles within the 130 days of simulations. For example, under the old inlet configuration, there was only 1.0% and 10.5% from the initial particles concentration released at R3 have successfully exited the lagoon throughout the simulation period of the southwest and northeast monsoons,

respectively (Figure 11a,b). Meanwhile, 93.5% and 80.0% of the R1 releases for the new inlet scenario were also retained in the Setiu Lagoon within the 130 days of simulation of the southwest and northeast monsoons, respectively (Figure 11a,b), making it difficult to quantify its residence time. However, despite the failure to estimate the residence time for some scenarios, the simulations have yielded spatial variability in residence times under different inlet configurations, providing evidence on how these scenarios may alter the particles concentration within the Setiu Lagoon.

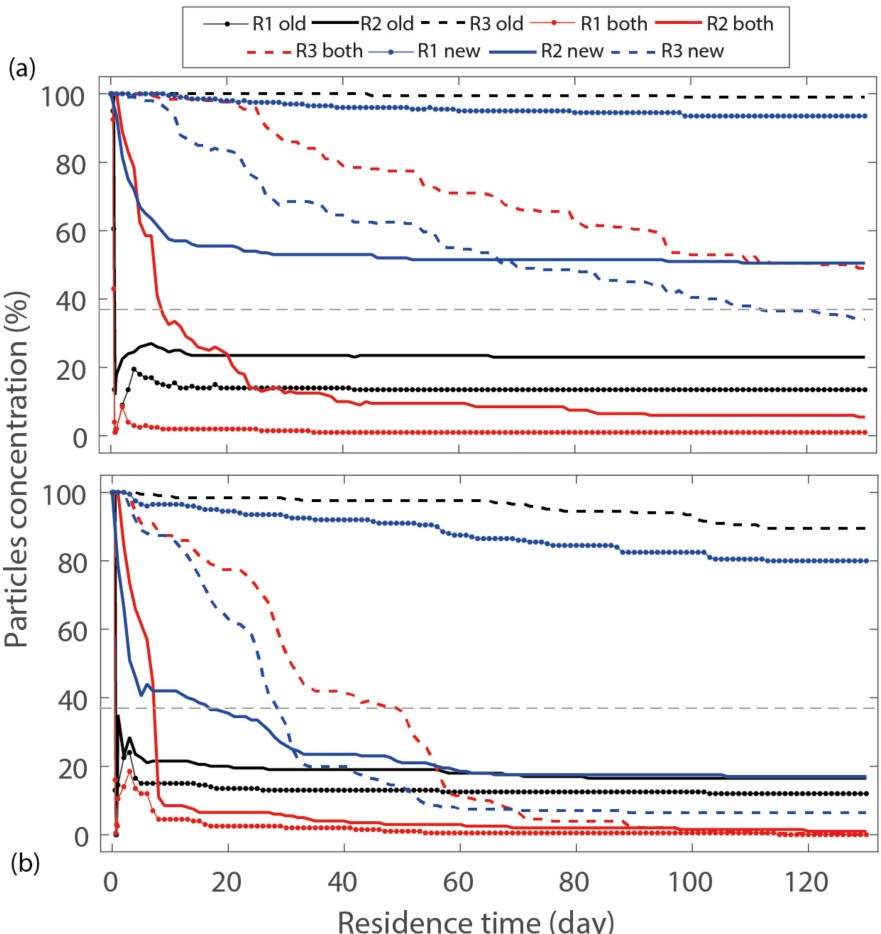

**Figure 11.** Particles concentration within the Setiu Lagoon under different inlet configurations released from R1, R2 and R3 during the (**a**) southwest and (**b**) northeast monsoons. The time when the particles concentration reaches 37% of its original values is presented in grey dash lines.

As summarized in Table 4, different inlet configurations in Setiu Lagoon have resulted in variation of residence time ranging from hours to days. Under the old inlet configuration, R1 exhibited the shortest residence time of 15 h during the southwest monsoon, followed by R2 (18 h) and R3 (>130 days; Figure 11a and Table 4). During the northeast monsoon, the residence time at R1 was 1.1 to 1.3 times shorter than the southwest monsoon (Figure 11b and Table 4). Comparing between different inlet scenarios, it seems like under the presence of both inlets, Setiu Lagoon had a good flushing efficiency represented by residence times of less than 50 days for all release locations and monsoon seasons (Figure 11 and Table 4). Although 37% particles concentration was not achieved by R3 releases for both inlets configuration during the southwest monsoon, it is noteworthy that only 49% of the initial tracer remained within the lagoon after 130 days, suggesting the estimation of residence time could be reached within the next additional days (Figure 11a and Table 4). Under the new inlet configuration, R1 displayed the longest residence time by eliminating only 6.5% and 20.0% of its initial concentration out of the Setiu Lagoon during the southwest

and northeast monsoons, respectively (Figure 11a,b). Meanwhile, the shortest residence time was recorded at R3, which were 112 days during the southwest monsoon and 29 days during the northeast monsoon (Figure 11 and Table 4).

**Table 4.** Residence time estimated at different release points under different inlet configurations in the Setiu Lagoon during the southwest and northeast monsoons.

| Inlet Configuration | Release Point | Monsoon | Residence Time |
|---|---|---|---|
| Old inlet | R1 | Southwest | 15 h |
| | | Northeast | 12 h |
| | R2 | Southwest | 18 h |
| | | Northeast | 15 h |
| | R3 | Southwest | >130 days |
| | | Northeast | >130 days |
| Both inlets | R1 | Southwest | 15 h |
| | | Northeast | 12 h |
| | R2 | Southwest | 9 days |
| | | Northeast | 8 days |
| | R3 | Southwest | >130 days |
| | | Northeast | 49 days |
| New inlet | R1 | Southwest | >130 days |
| | | Northeast | >130 days |
| | R2 | Southwest | >130 days |
| | | Northeast | 17 days |
| | R3 | Southwest | 112 days |
| | | Northeast | 29 days |

## 4. Discussion

This study is the first attempt to quantify the particles transport and its residence time from the past and current configurations of a shallow and narrow Setiu Lagoon using combined hydrodynamic and particle tracking models. Although the work presented here is a case study based, the results obtained could be beneficial to provide estimation of changes in water current flow and residence time for other lagoon systems with similar characteristics. This study is also extremely important to understand the changes of water flow within the Setiu Lagoon influenced by the relocation of the inlet, while evaluating its sensitivity to pollution. Although this area has a relatively complex bathymetry, the accuracy assessment showed in Tables 1 and 2 indicated that the hydrodynamic modeling of Setiu Wetland has been successfully validated. Furthermore, high simulation skills for all parameters, represented by an index of agreement that is close to 1 proved the capability of the simulation to produce realistic hydrodynamic conditions similar to observations (Tables 1 and 2).

With no freshwater input considered, the hydrodynamic numerical experiments suggest that the water current in Setiu Lagoon followed the neritic incursion (flood) and excursion (ebb) of the water level of the sea, producing a back-and-forth motion in water movement (Figure 4). Although tidal current plays a great role in shaping water flow inside Setiu Lagoon, its effect was limited to the inlet and central areas, in other words to different locations of the river mouth. At the northern and southern sectors, the shallowness of the channels has modified water current strength. This was evidenced through the intensification of the ebb and flood currents that took place in the inlet and central areas, while the water flow was weakened as it approached the northern and southern parts of the lagoon (Figure 4). The effect of shallow bathymetry in modulating the intensity of water current flow is aligned with other works on lagoons and estuaries that include [36,37]. Additionally, winds were believed to interact with tides in the coastal lagoon, creating a residual circulation in this area [17].

Hypothetically, the spatially and temporally varying tidal currents were expected to shape the distribution of the particles within Setiu Lagoon. This statement is supported

since the particle trajectories presented in Figures 5–10 were found to be in agreement with the tidal flow patterns (Figure 4) that exhibited back-and-forth movements, linked to the ebb and flood tides. Moreover, in all inlet configurations, particles that resided in the central part of the lagoon were easily transported northward and southward, or out of the lagoon, corresponding to strong tidal current strength in this area (Figures 5–10). Meanwhile, elongated stay of the particles in the northern and southern parts was predominantly associated with weaker tidal flow due to the presence of narrow channels (Figures 5–10).

When comparing the particle transport and residence time under different inlet configurations, the simulation indicates that there were substantial differences between the release points. Under the old inlet configuration, the R3 releases showed much higher residence time than R1 and R2 (Figures 5 and 6), with substantial trapping of particles within the southern sector, resulting in a residence time in excess of 130 days (Figure 11 and Table 4). Such a high residence time in this area suggests the potential for water quality degradation associated with domestic waste input from the nearby area of Penarik town [17,18,22,23]. In comparison, the movement of particles released from R1 and R2 was faster (Figures 5 and 6), resulting in a residence time of less than 20 h (Figure 11 and Table 4). The differences between the flushing rate of particles and residence time between the release locations were due to asymmetries of tidal current strength between areas, as well as proximity to the inlet [38,39].

Under both inlets configuration, the transport of particles out of the lagoon was improved, and the residence time lies between 12 h to 49 days (Figure 11 and Table 4). In this scenario, all release points are located close to the inlets (Figure 2b), where rapid water exchange occurs. With close proximity to the river mouth, residence time could be reduced since dilution with the incoming water was rapid [33]. Furthermore, with the presence of two inlets, each inlet no longer functions in isolation, instead forming a flow field between the old and new inlets [40]. This was evidenced from Figure 4e,g, in which during the flood, the tidal currents emanating from the new inlet area are directed northward with increased strength, while the currents emanating from the old inlet are directed southward unchanged, creating a flow-through system.

Under the new inlet configuration, R1 that is situated relatively far from the river mouth (6.8 km) remained in a stagnant state, with a residence time of more than 130 days (Figure 11 and Table 4). On account of its longest distance from the inlet, the water renewal in the northern part was time-consuming, and hence increased the flushing time for the particles. Moreover, strong flood jet originating from the new inlet (Figure 4j,l) pushed the R1 releases further northward, making it difficult for the ebb tidal currents to transport these particles back towards the river mouth. In addition, the presence of vegetated islands within Setiu Lagoon could act as obstacles, increasing retention and lengthening the residence time [1,18,41]. Considering the location of R1 that is characterized by dense aquaculture activities, longer residence time in this area suggests the possibility of water quality problems, as reported in [16,17,22]. However, relative to R1, the particles released at R2 and R3 exhibited a median residence time, ranging between 17 to 112 days (Figure 11 and Table 4), particularly due to their short distance from the river mouth and the good flushing efficiency of the area.

Within the present work, the northeast monsoon notably resulted in shorter residence time relative to the southwest monsoon for all inlet configurations (Figure 11 and Table 4). Without the quantification of river discharge, the difference in residence time estimates between different monsoon seasons provides evidence on how seasonal variations may shape the temporal variability in the distribution of tidal flow and particles transport within Setiu Lagoon. On average, the tidal current during the northeast monsoon was in the order of 1 to 2 times stronger than during the southwest monsoon (data not shown). Such increased values of tidal currents during the northeast monsoon could promote an increase in the flow velocities within the lagoon, and hence produce a flushing twice as fast as that of the southwest monsoon. Furthermore, strong average northeast monsoon winds of 4.6 m/s compared to the southwest monsoon (2.2 m/s) were strongly believed to expedite

the transport of the exited particles out of the ocean boundary, thus preventing a large number of re-entrant particles. Moreover, despite it shorter distance to the river mouth, R2 releases failed to reach its 37% particle concentration during the southwest monsoon under both inlets and the new inlet configurations (Figure 11 and Table 4), strengthening the role of monsoonal forces in shaping the residence time distribution within Setiu Lagoon. This seasonal influence on the residence time is in agreement with studies conducted in other shallow coastal water bodies, for example, Gulf of Kachchh [33], Pearl River Estuary [42], and Chilika lagoon [43].

## 5. Conclusions

A two-dimensional coupled hydrodynamic and particle tracking model was applied to Setiu Lagoon, which is a shallow and narrow coastal lagoon that experienced inlet relocation due to strong monsoonal forces. Using an implementation of a conservative tracer simulation, this study aimed to determine the particle transport within the lagoon, and its residence time under three different inlet configurations. The hydrodynamic simulations showed substantial variability in the distribution of tidal flows throughout Setiu Lagoon, with higher water current intensity recorded at the inlet and central area relative to the northern and southern parts. The simulated residence time assessment was in accordance with the distribution patterns of tidal currents, in which higher particle flushing was recorded in the central area than in the northern and southern parts. Substantial spatially and temporally varying residence time in Setiu Lagoon suggested asymmetry tidal currents as the major influencing factor for particle distribution within the lagoon, while close proximity to the river mouth was the key element for residence time. In addition, strong northeast monsoon forces have increased the flushing capacity within the lagoon, resulting in shorter residence time. On account of different inlet configurations, simulations on the combined inlets scenario yielded the shortest residence time in Setiu Lagoon, suggesting its competency in removing pollutants from the system. The present study recommends extended periods of particle tracking simulation to avoid uncertainties pertaining to the estimation of residence time. The present findings are extremely beneficial to local authorities for possible conservation and management efforts of Setiu Lagoon and for its ecological and economic importance.

**Author Contributions:** Conceptualization, Z.Z. and M.F.A.; methodology, Z.Z. and M.F.A.; software, Z.Z. and M.F.A.; validation, Z.Z. and M.F.A.; formal analysis, Z.Z.; investigation, Z.Z.; data curation, Z.Z.; writing—original draft, Z.Z.; writing—review and editing, M.F.A.; visualization, Z.Z. and M.F.A.; supervision, M.F.A. All authors have read and agreed to the published version of the manuscript.

**Funding:** This research has been supported by the Institute of Oceanography and Environment, Universiti Malaysia Terengganu through the Higher Institution Centre of Excellence (HICOE) program. The research is part of first author postgraduate work that was supported by the Tunku Chancellor Scholarship under the Universiti Malaysia Terengganu.

**Institutional Review Board Statement:** Not applicable.

**Informed Consent Statement:** Not applicable.

**Data Availability Statement:** Not applicable.

**Acknowledgments:** The authors would like to deliver special thanks to every colleague who helped us in carrying out both field and lab-based observations successfully.

**Conflicts of Interest:** The authors declare no conflict of interest.

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
