# Peer review of "The Effects of Different Inlet Configurations on Particles Transport and Residence Time in a Shallow and Narrow Coastal Lagoon: A Numerical Based Investigation"

_water, doi:10.3390/w14091333_

Round 1
Reviewer 1 Report
Please see the attached report for comments.

Reviewer 2 Report
General Comments:
The manuscript titled “The Effects of Different Inlet Configurations on Particles Transport and Residence Time in a Shallow and Narrow Coastal Lagoon: A Numerical Based Investigation” by Z. Zainol & M. F. Akhir, investigates the effects inlet configurations on hydrodynamics of a multi-inlet coastal lagoon by means of numerical modelling. Specifically, the authors assess lagrangian transport and residence time. The topic is interesting to the journal and, in particular, it would find readers interested in coastal hydrodynamics. However, there are several aspects that need to be detailed before the manuscript can be accepted for publication.
Specific comments:
- Provide details of water levels and currents in the region of study, i.e. tide type, spring-neap ranges, residual components seasonal variations. A timeseries showing the total, tide and residual of water levels and currents will be useful to help the reader develop a conceptual model of the hydrodynamics and forcings of the region.
- Wind forcing was turned off during the simulations but in the introduction is mentioned that plays an important role in sediment transport. In this particular study, do the authors consider that the wind plays an important role on particles transport and residence time? Please, give details.
- In the introduction, the authors reference several publications about Ria Formosa multi-inlet lagoon. Even though they are interesting and appropriate for this study, consider to add a more recent publication about anthropogenic impacts on Ria Formosa multi-inlet system, e.g. https://doi.org/10.1016/j.apenergy.2018.09.204
- Provide details of the mesh used to discretise the model domain, i.e. number of elements, min./max. size of elements. A plot showing a general view of the mesh, as well as a zoom over the refined zones.
- Show timeseries and scatter plots comparing water levels and current speeds of model vs observations.
Round 2
Reviewer 1 Report
Authors have satisfactorily addressed my comments, and the manuscript in the present form can be accepted for publication.
Reviewer 2 Report
I'm pleased with the revisions made by the authors, therefore, I recommend the manuscript for publication.